# Effect of Short-Chain Fatty Acids and Polyunsaturated Fatty Acids on Metabolites in H460 Lung Cancer Cells

**DOI:** 10.3390/molecules28052357

**Published:** 2023-03-03

**Authors:** Tianxiao Zhou, Kaige Yang, Jin Huang, Wenchang Fu, Chao Yan, Yan Wang

**Affiliations:** School of Pharmacy, Shanghai Jiao Tong University, Shanghai 200240, China

**Keywords:** H460 lung cancer cells, short-chain fatty acids, polyunsaturated fatty acids, untargeted metabolism, targeted metabolism

## Abstract

Lung cancer is the most common primary malignant lung tumor. However, the etiology of lung cancer is still unclear. Fatty acids include short-chain fatty acids (SCFAs) and polyunsaturated fatty acids (PUFAs) as essential components of lipids. SCFAs can enter the nucleus of cancer cells, inhibit histone deacetylase activity, and upregulate histone acetylation and crotonylation. Meanwhile, PUFAs can inhibit lung cancer cells. Moreover, they also play an essential role in inhibiting migration and invasion. However, the mechanisms and different effects of SCFAs and PUFAs on lung cancer remain unclear. Sodium acetate, butyrate, linoleic acid, and linolenic acid were selected to treat H460 lung cancer cells. Through untargeted metabonomics, it was observed that the differential metabolites were concentrated in energy metabolites, phospholipids, and bile acids. Then, targeted metabonomics was conducted for these three target types. Three LC-MS/MS methods were established for 71 compounds, including energy metabolites, phospholipids, and bile acids. The subsequent methodology validation results were used to verify the validity of the method. The targeted metabonomics results show that, in H460 lung cancer cells incubated with linolenic acid and linoleic acid, while the content of PCs increased significantly, the content of Lyso PCs decreased significantly. This demonstrates that there are significant changes in LCAT content before and after administration. Through subsequent WB and RT-PCR experiments, the result was verified. We demonstrated a substantial metabolic disparity between the dosing and control groups, further verifying the reliability of the method.

## 1. Introduction

Lung cancer is a malignant tumor from the bronchial mucosa or glands within the lungs and has the fastest-growing morbidity and mortality. It is one of the most threatening malignancies to human health and life [1,2]. More than 1.6 million people have lung cancer annually, which is a deadly form of cancer. In particular, non-small cell lung cancer (NSCLC) is quite common, accounting for 80% of all lung cancer cases.

Fatty acids include short-chain fatty acids (SCFAs) and polyunsaturated fatty acids (PUFAs) as essential components of lipids. Short-chain fatty acids are lipids composed of two to six carbon atoms. They include acetate, propionate, and butyrate, mainly produced by the digestion of dietary fiber by intestinal microorganisms [3,4,5]. SCFAs can be recognized by G-protein-coupled receptors (GPCRs) on the surface of cancer cells. In addition, SCFAs can also enter the nucleus of cancer cells [4], inhibit histone deacetylase activity, upregulate histone acetylation and crotonylation, and play a direct anti-cancer effect due to their small molecular weight [6,7]. SCFAs can also exert indirect anti-cancer effects by regulating immune cells [8].

Polyunsaturated fatty acids are straight-chain fatty acids with two or more double bonds and have 18–22 carbon atoms in their chain. They are generally divided into omega-3 and omega-6. Omega-3 polyunsaturated fatty acids can provide energy and prevent and treat diseases as one of the essential nutrients for the human body [9]. Recent studies have found that omega-3 PUFAs (viz., eicosapentaenoic acid (DHA), linolenic acid, and docosahexaenoic acid (EPA)) can improve the chemotherapy sensitivity of tumor cells. Moreover, they reduce the side effects of chemotherapy and protect target tissues without any adverse effects on non-target tissues. Many epidemiological survey results show that the incidence of liver, breast, prostate, and colon cancer in people with high omega-3 PUFAs is decreased [10,11]. Omega-3 PUFAs inhibit the growth of lung cancer cells and play a vital role in inhibiting their migration and invasion. The omega-6 family includes linoleic acid and arachidonic acid (ARA), precursors of pro-inflammatory oxylipins [12]. There is clinical evidence suggesting the effect of omega-6 PUFAs on the progression of lung cancer [13,14,15]. For instance, Liu et al. revealed that lung adenocarcinoma and squamous cell carcinoma patients had higher levels of free fatty acids (e.g., ARA and linoleic acid) and their hydroxyeicosapentaenoic acid (HETE) metabolites compared with the non-cancer control group, proposing these as possible markers [12]. However, for H460 cells, there are few studies on the administration of SCFAs and PUFAs, and the mechanism and different effects of SCFAs and PUFAs on lung cancer are still unclear.

Metabonomics is a research method imitating the research ideas of genomics and proteomics, conducting a quantitative analysis of all metabolites in the organism and searching for the relationship between metabolites and physiological and pathological changes [16,17,18,19,20,21,22,23,24,25]. It is a component of systems biology. The objects are primarily small molecular substances with a relative molecular weight of less than 1000. Advanced analytical detection technology combined with computational analysis methods, including pattern recognition and expert system, is the basis of metabonomics research. Among these, metabolomics involves untargeted and targeted metabolomics. Untargeted metabolomics systematically and comprehensively analyzes the whole metabolome based on limited relevant research and background knowledge, obtains several metabolite data, and processes them to identify the differential metabolite [26]. Presently, untargeted metabonomic analysis is widely used in biomarker discovery, disease diagnosis, and mechanism research. Thus, it provides new ideas and directions to understand the disease mechanism. Targeted metabonomics analyzes and studies only a limited number of metabolites associated with biological events based on the principles and concepts of metabonomics [16,27]. Usually, targeted metabonomics is used for systematic confirmation after finding differential metabolites through untargeted metabonomics [28,29]. The newly developed glycomics and lipomics also belong to targeted metabonomics. Targeted metabonomics is precise in analysis and complementary to untargeted metabonomics. Metabonomics is indispensable for studying the fatty acid mechanism in H460 lung cancer cells.

In this study, SCFAs involved sodium acetate and sodium butyrate, and PUFAs included linoleic acid (omega-6) and linolenic acid (omega-3) for treating H460 lung cancer cells. Untargeted metabonomics was used to find essential metabonomics enrichment pathways. Depending on the result of untargeted metabonomics, a method was established for simultaneously quantifying 71 metabolites involved in energy metabolites, bile acids, and phospholipids. It also covers glycerophospholipid metabolism, primary bile acid biosynthesis, the TCA cycle, and the ATP pathway. Then, targeted metabonomics experiments were conducted on these target metabolites. The metabolite changes are very different in H460 lung cancer cells incubated with SCFAs and PUFAs. Primarily, the content of PCs increased significantly and Lyso PCs decreased significantly when H460 lung cancer cells were incubated with linolenic acid and linoleic acid. Further analysis of the KEGG pathway indicated that the above differential metabolites could be due to the changes in LCAT expression. Thus, we verified the previous experimental metabonomics results by utilizing the WB and qPCR experiments for LCAT. We also clarified the different effects of four fatty acids on H460 lung cancer cells.

## 2. Results

### 2.1. CCK8 Experiment

The first step in studying cell metabonomics is the cell proliferation experiment of the fatty acid and control groups. To ensure that the dose does not affect cell proliferation and that further metabonomic analysis can be carried out, we conducted a CCK8 experiment on the fatty acid and control groups. The experimental results are shown in Appendix A. When the concentration of the four fatty acid groups is 500 μM, cell proliferation is unaffected. However, cell proliferation was affected to a certain extent when the linolenic acid concentration continued to increase. Therefore, 500 μM was selected as the fatty acid concentration of the four fatty acid groups.

### 2.2. Untargeted Metabolism

We used hydrophilic chromatography–mass spectrometry (HILIC-MS/MS) and reverse-phase chromatography–mass spectrometry (RPLC-MS/MS) to detect more metabolites in untargeted metabonomics. Both used positive and negative ion modes simultaneously to detect polar and non-polar molecules in cell samples. Multi-dimensional statistical analysis of data collected from untargeted metabolism is depicted in Figure 1A and Figure 2A. From the figures, QC samples, the control group, and samples in the four fatty acids administration groups are in similar geometric positions. Therefore, these samples were naturally clustered, with small differences between groups, and the overall distribution was good. There was no significant singular value to be removed, and they were uniformly distributed within 95% of the confidence interval (different color areas). QC sample aggregation established the stability of the instrument from the side, consistent with the previous validation results. In PCA, there is a certain degree of separation between groups. OPLS-DA was used to explore the metabolic difference between the fatty acid and the control groups. The fatty acid and the control groups were separated in the OPLS-DA score chart (Figure 1B–E and Figure 2B–E). The differences between groups were amplified after supervised target classification and discrimination, and a more noticeable trend separation was observed. It is necessary to evaluate the quality of the discriminant models. The results indicated that the interpretation rate (R2Y) of each model to the original data and the prediction rate (Q2) to the grouping were very good.

The ultimate purpose of the reliable model was to screen the heterologous variables from massive metabolic data and extract information with biochemical significance. Therefore, screening differential metabolites is a crucial step in statistical processing. We selected a multi-standard evaluation method to improve the reliability and accuracy of screening.

First, the variables with VIP > 1 were selected and had statistical significance based on the multi-dimensional statistical model discrimination (all variables are considered). Secondly, it was assisted by single-dimension statistical analysis (evaluation of a single variable). We adopted the Student’s *t*-test method without assuming data distribution in advance since the number of samples was small and the overall standard deviation was unknown. The *p*-value was obtained using Student’s *t*-test, which determined the significance of the difference between the two groups. The significance level was 0.05. The fold change ratio (FC) was obtained by dividing the average value of the variables in the administration group (before normalization, the ratio of the average values of paired data was selected) by the average value of the healthy group. This was used to make a macro performance comparison of the variables in the two groups. When FC > 1, the mean value of the fatty acid group was higher than the control group, and the variable was upregulated in the fatty acid group. When FC < 1, the mean value of the fatty acid group was lower than the control group, and the variable was downregulated in the fatty acid group. The absolute value FC > 1.5 or FC < 0.67 was selected as the screening criteria for statistics. Through the above screening criteria, and depending on the differential expression multiples and significance results, the volcano map (see Figure 2A–H) was drawn for screening the differential metabolites between the two sample groups. Based on the volcano plot results, 50 compounds were selected with the largest metabolic difference in untargeted metabolism between the heat map and correlation heat map.

Based on the ratio of the average response intensity of the TOP50 differential metabolite of HILIC and RP-18 in the fatty acid and the control group samples, two heatmaps were drawn (Figure 3A for HILIC, Figure 3B for RP-18). It can be seen that phospholipids, energy metabolites, and bile acids in the administration group showed significant changes. Moreover, the upregulation of energy metabolites in the short-chain fatty acid administration group was more prominent, and the increase and decrease in phospholipids in the long-chain fatty acid group were evident.

The correlation analysis results of metabolites are demonstrated in Figure 3C,D. From the results, PCs were negatively correlated with Lyso PCs, ADP in energy metabolites was positively correlated with PCs, and fumaric acid and pyruvic acid were negatively correlated with PCs. The results also revealed that there were many phospholipids and energy metabolites in untargeted metabolism TOP50, including PS (22:6), Lyso PC (20:5), PI (20:3), etc., in phospholipids, and ADP, CDP, Glyceraldehyde 3-phosphate, etc., in energy metabolites. After analyzing the correlation of these differential metabolites using untargeted metabolism, we conducted an enrichment analysis of the KEGG pathway. As shown in Figure 4, many pathways of four fatty acid changes were involved in the citrate cycle (TCA cycle), gluconeogenesis, and glycerophospholipid metabolism.

The data analysis of untargeted metabonomics showed that the different metabolites before and after administration were mainly concentrated in energy metabolites, phospholipids, and bile acids. Therefore, a follow-up targeted metabonomic analysis was conducted for these three types of substances.

### 2.3. Building and Optimizing the Analytical Method for Targeted Metabolome Detection

#### 2.3.1. Pretreatment Optimization

The type and polarity of the extraction system influence the extraction efficiency of target substances. Eight extraction solvent systems were designed, which were chloroform: water (1:1), methanol: water (1:1), chloroform: acetone: water (1:1:1), chloroform: ethanol: water (1:1:1), chloroform: methanol: water (1:1:1), chloroform: methanol: water (2:1:1), chloroform: methanol: water (1:1:2), chloroform: methanol: water (1:1:2), and chloroform: methanol: water (1:2:1). The above proportions were the volume ratios. As shown in Appendix A, most internal standards in chloroform: methanol: water (1:2:1) extraction recovery rates were in the 90–110% range, better than other extraction systems. The extraction system of chloroform: methanol: water (1:2:1) was subsequently utilized for pretreatment.

#### 2.3.2. LC-MS Method Optimization for Targeted Metabolism

The Waters BEH amide column (100 mm × 4.6 mm i.d., 3.5 μm) was selected for the chromatographic analysis of energy metabolites. Chromatographic separation was carried out, followed by the optimization of the pH of the mobile phase and the added concentration of ammonium acetate. The peak shape and response intensity were optimal when pH = 9.0 and the ammonium acetate concentration was 10 mM. Finally, we used pH, the additive concentration of mobile phase with pH = 9.0, and the ammonium acetate content of 10mM as energy metabolite for analysis.

We selected Waters BEH C18 column (100 mm × 4.6 mm i.d., 3.5 μm) for the chromatographic analysis of phospholipid and bile acid. The chromatographic separation was performed, and the overall analysis time was compressed to 20 min by raising the column temperature to 50 °C. Simultaneously, 5 mM of ammonium formate was added to the aqueous phase to enhance the peak shape of some target compounds. All the optimized TIC chromatograms are represented in Appendix A. The energy metabolite chromatograms are shown in Appendix A, and the phospholipids and bile acid chromatograms for ESI+ mode are depicted in Appendix A. Chromatograms of phospholipids and bile acids for ESI- mode are demonstrated in Appendix A.

As shown in Appendix A, parent and daughter ions for different target compounds were determined, and the DP, EP, CE, and CXP values were optimized.

#### 2.3.3. Targeted Methodological Validation

The method validation results are depicted in Table 1. The LODs were in the 0.001–0.766 ng/μL range, and the LOQs were in the 0.003–2.553 ng/μL range. Since the linear regression coefficients (R^2^) of most metabolites are above 0.99, the method possessed good linearity. The intraday and intraday RSD% precision of most metabolites was within ± 20%, reflecting good method precision.

The recovery rate and matrix effect of cells are indicated in Table 2. The recoveries and matrix effect of internal standards in cell samples were more significant than 81.9% and 87.6%. Therefore, this established method is suitable for quantitatively analyzing target metabolites in cell samples since the recovery rate and matrix effect of most metabolites in the cell matrix are more than 80%.

MQC of cell samples was used for the stability study. Under short-term and long-term storage conditions, the peak area ratio was more than 83.6%. Therefore, the target metabolite was stable when stored at 4 °C for 24 h or −20 °C for two months (Table 3).

### 2.4. Targeted Metabolism

The final variable (metabolite concentration) can directly reflect the content of a single component in a sample. Moreover, the variable strength is not on a unified scale, significantly affecting the statistical results. Therefore, the Par scaling method was used to increase the comparability of each variable across different samples. The specific operation was to divide the variable by the square of its standard deviation, and, based on not amplifying noise interference, simultaneously, the contributions of high and low-content metabolites were also considered. Appendix A indicates the comparison differences before and after data standardization.

The quantitative results were imported into the SIMCA software for multivariate analysis. PCA demonstrated that all the samples were clustered in the PCA chart (within a 95% confidence interval) (see Figure 5A), and it is not necessary to exclude the extreme outliers. The method has good repeatability due to its good intra-group aggregation. We also observed that the control group was close to the sodium acetate group, and the linoleic acid group was nearby the linolenic acid group. To further compare the difference between the administration and control groups, we conducted an OPLS-DA analysis.

OPLS-DA was used to explore the metabolic difference between the fatty acid and control groups. The fatty acid and control groups were separated in OPLS-DA score charts (Figure 5B–E). The OPLS-DA model parameters (R2X = 0.598, R2Y = 0.993, Q2 = 0.991; R2X = 0.717, R2Y = 0.991, Q2 = 0.993; R2X = 0.814, R2Y = 0.999, Q2 = 0.996; R2X = 0.734, R2Y = 0.999, Q2 = 0.994) established that the model has an excellent fitting and prediction ability. We also observed significant differences between the four fatty acid and control groups. We subsequently drew and analyzed the volcano plot to screen out the different metabolites between the groups.

The volcano plot of targeted metabolism is depicted in Figure 5F–I. Based on the analysis, VIP > 1.0, *p*-value < 0.05 (Student’s *t*-test), and FC >1.5 were selected. The different metabolites of the four fatty acid groups compared with the control group were found by drawing the volcano plot. A follow-up analysis was also carried out. The heat map and correlation analysis of the screened differential metabolites were also determined.

The heat map of targeted metabolism is shown in Figure 6A. It can be observed that these target compounds possess significant changes in different administration groups. For instance, compared to the control group, D-Fructose-1,6-Diphosphate, and α-ketoglutaric acid downregulated in the SCFA group but upregulated in the PUFA group; 16:0–18:1 PC, 16:0–18:1 PA upregulated in the SCFA group, but downregulated in the PUFA group.

We noticed that in the linoleic acid and linolenic acid fatty acid group, all the PCs and Lyso PCs we targeted to detect had a specific change rule. Lyso PCs increase with the decline of PCs.

The correlation analysis results of metabolites are represented in Figure 6B. Moreover, all PCs were positively associated with each other and negatively correlated with all Lyso PCs. The results of the heatmap are also verified.

### 2.5. Effects of LCAT on Metabolome Changes and Biological Validation

It can be seen from the above-targeted metabolism statistical results and box plots (Figure 7) that the changing trend of PCs and Lyso PCs in the control and the administration groups had a particular rule among all the target compounds. Especially in the two long-chain unsaturated fatty acid treatment groups, viz., the linoleic acid and linolenic acid treatment group, Lyso PCs increased simultaneously with their decrease. According to the KEGG metabolic pathway map, the above changes could be due to the rise of LCAT. Therefore, we designed WB and qPCR experiments to verify the hypothesis.

#### 2.5.1. mRNA Expression of LCAT

We tested whether four fatty acids altered the gene expression of LCAT (Figure 8A). H460 lung cancer cells were exposed to 500 μM of sodium acetate, butyrate, linoleic acid, and linolenic acid. As shown in Figure 8A, LCAT expressions were differently affected by exposure to the four fatty acids. The expression level of LCAT was upregulated during linolenic acid treatment but downregulated when treated with sodium acetate, butyrate, and linoleic acid. These results suggested that fatty acids significantly and differently affected H460 lung cancer cells. To further study the effect of four fatty acids on LCAT expression, we conducted a WB experiment.

#### 2.5.2. Protein Content of LCAT

The qPCR results showed that the four fatty acids affected the gene transcription level of LCAT in varying degrees. H460 lung cancer cells were exposed to 500 μM of sodium acetate, sodium butyrate, linoleic acid, and linolenic acid to confirm the effect of four fatty acids on LCAT. Western blot results (Figure 8B–D) revealed that linolenic acid induced the strongest LCAT expression. The other three fatty acids generated no significant changes in LCAT expression.

## 3. Discussion

Lung cancer has the fastest-growing incidence rate and mortality, which seriously threatens human health and safety [1]. Based on the latest data released by the World Health Organization, lung cancer incidence rate and mortality rank first among all kinds of malignant tumors globally [2].

Lipid metabolism, especially fatty acid synthesis, is an essential cellular process that can convert nutrients into metabolic intermediates for membrane biosynthesis, energy storage, and signal molecule generation [30,31]. Lipid metabolism is a vital metabolic phenotype of cancer cells. Therefore, blocking the lipid supply in cancer cells will significantly impact the bioenergetics, membrane biosynthesis, and intracellular signal transduction processes of cancer cells [32]. Among them, short-chain and polyunsaturated fatty acids have a more significant impact on lung cancer. In a recent study, sodium butyrate treatment upregulated miR-3935, thereby inhibiting the growth and migration of A549 cells [33]. In addition, sodium butyrate inhibits the growth of lung and prostate cancer by regulating the p21 expression [34]. However, the metabolic effect of sodium butyrate on lung cancer, particularly on H460 lung cancer cells, is unclear. For other SCFAs, the metabolic effects on lung cancer are also inconclusive. Polyunsaturated fatty acids (PUFAs) are straight-chain fatty acids with two or more double bonds in their structure and a carbon chain length of 18~22 carbon atoms. It is divided into omega-3 and omega-6. Omega-3 PUFAs mainly include eicosapentaenoic acid (EPA) and docosahexaenoic acid (DHA), and omega-6 PUFAs primarily include linoleic acid and arachidonic acid. Omega-3 and omega-6 fatty acids are essential components of cell membranes and precursors of many biochemical reactions in vivo, including regulating blood pressure and inflammatory reaction. Linoleic acid (LA) in omega-6 and omega-3 fatty acids, viz., α- Linolenic acid (ALA), cannot be synthesized by the human body but is only generated through food intake. PUFAs can promote apoptosis and inhibit cell proliferation of many malignant tumors including breast, liver, and pancreatic tumors [9,11]. The possible mechanisms of the polyunsaturated fatty acid effect on cancer include influencing the composition and function of the biomembrane, controlling lipid peroxidation of tumor cells, inhibiting the expression and function of oncogene-encoded proteins, tumor angiogenesis, and cancer cell adhesion to endothelial cells.

Although the physiological effects of fatty acids have been studied, their mechanisms and pathways of action on lung cancer, and the differences between short-chain and polyunsaturated fatty acids, have not been analyzed. This study compared the effects of short-chain and polyunsaturated fatty acids on lung cancer cells via untargeted and targeted metabolic groups. First, the effects of different concentrations of sodium acetate, sodium butyrate, linoleic acid, and linolenic acid were investigated on the survival rate of lung cancer cells. As shown in Appendix A, exposure to 500 μM of sodium acetate, sodium butyrate, linoleic acid, and linolenic acid could not affect the survival of H460 cells. However, greater than 500 μM of linolenic acid had the highest cytotoxicity in H460 cells. Therefore, we selected 500 μM of fatty acids to treat lung cancer cells and investigate their impact on metabonomics in the follow-up experiment.

Untargeted metabolomics systematically identifies and analyzes the whole-life metabolite based on limited background knowledge, obtains a large amount of metabolite data, and identifies differential metabolites. Simultaneously, untargeted metabonomics is also an essential prerequisite for targeted metabonomics. We found that the metabonomic characteristics of H460 cells changed significantly after exposure to fatty acids using the untargeted analysis of the above four fatty acid and control groups. As shown in Figure 1A and Figure 2A, PCA results showed that both intra-group aggregation and inter-group separation were good. As shown in Figure 1B–E and 2B–E, OPLS-DA results showed noticeable differences between the fatty acid and control groups. We have drawn volcano plots to analyze the distribution of different metabolites and their relationship between groups (Figure 1F–I and Figure 2F–I). VIP > 1.0, *p*-value < 0.05 (Student’s *t*-test), and FC value > 1.5 times are selected to identify different metabolites between the fatty acid and the control groups. There were a lot of different metabolites in the volcano plots compared with the control group. We selected 50 representative differential metabolites to determine heatmaps and correlation analysis heatmaps (Figure 3). The heatmap results confirmed that the intra-group aggregation of our untargeted metabonomics is excellent, and some differences exist between groups. The correlation analysis heatmaps revealed that PCs were negatively correlated with Lyso PCs, ADP in energy metabolite was positively associated with PCs, and fumaric acid and pyruvate were negatively correlated with PCs, etc.

The statistical analysis of untargeted metabonomics showed that the different metabolites between the fatty acid and the control groups were mainly concentrated in energy metabolites, phospholipids, and bile acids. Then, we carried out targeted metabonomics research. We first establish three LC-MS/MS methods for 71 compounds, including energy metabolites, phospholipids, and bile acids. The validity of the method was verified by the subsequent validation results.

The interference of four fatty acid treatments on the cell metabolism group was substantial. There were 17 and 20 different metabolites in the sodium acetate and butyrate groups and 19 and 23 in the linoleic and linolenic acid groups, respectively. Moreover, the energy metabolite, phospholipid, and bile acid were significantly changed. For the two groups of short-chain fatty acid administration and two polyunsaturated fatty acid administration groups, we continued to perform OPLS-DA analysis. As shown in Figure 9, the results showed significant differences between the short-chain fatty acid administration group and the polyunsaturated fatty acid group. The changes in metabolites are related to several metabolic pathways, which may reveal different potential mechanisms of fatty acid administration and metabolites. According to the results of the metabolic pathway analysis, four fatty acid pathways are involved in the citrate cycle (TCA cycle), gluconeogenesis, and glycerophospholipid metabolism. The sodium butyrate group in particular had noticeable metabolic changes in the short-chain fatty acid group. Figure 10A shows that PCs and PAs are downregulated, indicating that PLD expression is upregulated. While PCs are downregulated, Lyso PCs are also downregulated, which may indicate that the expression of LCAT or PLA is upregulated. When PCs are downregulated, PSs are upregulated, indicating the upregulation of PTDSS1 expression. While Pas are downregulated, Pes are upregulated, which may indicate that PLD expression is also upregulated. While TCA and GCA are upregulated, CA is also upregulated, which may indicate that the expression of CGH is downregulated. The linolenic acid group had noticeable metabolic changes in the polyunsaturated fatty acid group. As shown in Figure 10B, Lyso PCs are upregulated, while PCs are downregulated, indicating the upregulation of LCAT or PLA expression. While PA is downregulated, PC is also downregulated, and PE remains unchanged, which may indicate that the expression of PLD is upregulated. While PA is downregulated, PG is also downregulated, which may indicate that the expression of CDS and PGS1 is upregulated. When PCs are downregulated, PSs are also downregulated, which may indicate that the expression of PTDSS1 is upregulated. While TCA and GCA are upregulated, CA is upregulated, which may indicate that the expression of CGH is downregulated. Phosphoenolpyruvate and pyruvate were downregulated, indicating the upregulation of PKLR expression or PKM activity. We quantified phospholipid concentrations and observed that linoleic acid and linolenic acid led to significant changes in PCs and Lyso PCs levels. In the two long-chain unsaturated fatty acid groups, Lyso PCs increased with the decrease in PCs. Depending on the KEGG metabolic pathway map analysis, this altered rule may be due to the change of LCAT expression after administering long-chain unsaturated fatty acids.

Glycolysis is a ubiquitous pathway for glucose degradation. This process provides a certain amount of energy to the body, and the intermediate products provide raw materials for biosynthesis. The oxidative phosphorylation process is a coupling reaction to release energy from substances due to body oxidation while supplying adenosine diphosphate (ADP) and phosphorus to synthesize adenosine triphosphate (ATP) [35,36,37]. In mammalian cells, oxidative phosphorylation synthesizes the energy cells need [38,39]. The tricarboxylic acid cycle (TCA) is an essential metabolic pathway in the organism, the final metabolic pathway of the three major nutrients, and the hub of the three major nutrient metabolic links [40]. Tumor cells especially generate energy. Healthy cells release a lot of energy depending on mitochondrial oxidized carbohydrate molecules. In contrast, most tumor cells provide energy via glycolysis with relatively low productivity, called the Warburg effect. Hexokinase 2 (HK2) is the first enzyme that catalyzes hexose phosphorylation and the glycolysis pathway [41,42]. It is also the rate-limiting enzyme of the Warburg effect. HK2 enhances the proliferation of tumor cells, inhibits cell apoptosis, and promotes cell invasion and metastasis. It is crucial for the rapid growth of tumor metabolism [43,44]. Phospholipids (PLs) are the main components of cell membranes and plasma lipoproteins and are essential bioactive lipids [16,27,45]. They participate in various physiological processes, such as cell proliferation, survival, apoptosis, cytoskeleton construction, and pathophysiological processes, such as inflammation, atherosclerosis, and cancer [46,47,48,49,50,51,52,53]. Bile acid is an essential component of bile and plays a vital role in fat metabolism[54]. Bile acid can activate the Farni-like X receptor (FXR), which is crucial in regulating bile acid synthesis, secretion, and lipid and glucose metabolism in the liver [55]. Therefore, rapid and accurate quantification of metabolites is valuable and urgent for exploring the pathological mechanism of these diseases and screening potential biomarkers.

Phospholipids are the main component of cell membranes and are crucial for establishing barriers protecting cells from the surrounding environment and separating and controlling many cell processes. These molecules or their derivatives act as essential signals to regulate critical cellular reactions or play special roles, including lung surfactants. The length and unsaturation of these fatty acyl chains are different since phospholipids can have different head groups and are usually composed of multiple fatty acyl chains. Therefore, phospholipids have a wide variety because each species has unique properties, which can affect the folding, structure, and function of membrane proteins [50]. The specific lipid composition of the membrane determines its physical and functional characteristics with great biological significance. Several changes in phospholipid metabolism were observed in cancer cells. It has been recognized that cancer cells require more cell membranes to proliferate. Therefore, they synthesize fatty acids used as building blocks of phospholipids. This is achieved by significant overexpression and activation of key lipogenic enzymes, such as fatty acid synthase. In addition, enzymes involved in fatty acyl chain metabolism, hydrolysis, and remodeling, such as stearyl coenzyme A desaturase and several phospholipases, are abnormally expressed in cancer tissues. In addition, some of these enzymes are affected by non-small cell lung cancer.

Lecithin–cholesterol acyltransferase (LCAT) is from a family of crucial lipid metabolism enzymes with no structural characteristics. It is responsible for the reverse transport of cholesterol over the lung surface [56,57,58,59,60]. The LCAT structure indicates the molecular basis of most human diseases with known LCAT missense mutations. It paves the way for the rational development of new therapies for LCAT deficiency, atherosclerosis, and acute coronary syndrome [58,61,62,63,64,65,66,67,68,69,70]. LCAT is a potential biomarker for various cancers, including ovarian cancer [71], breast cancer [72], colorectal cancer [73], and liver cancer [74]. However, there has been limited study of LCAT in lung and lung cancer cells.

Lecithin–cholesterol acyltransferase (LCAT) is a 416 amino acid glycoprotein. It transesterifies sn-2 fatty acids of phospholipid molecules into a 3- α- Hydroxyl cholesterol group into two products, viz., lysophosphatidylcholine (Lyso PC) and cholesterol ester (CE) [56,57,62]. This enzyme has a role in the lipid–water interface of lipoprotein particles. It is associated with lipoprotein particles, such as plasma HDL, LDL, newborn discoid HDL, and recombinant HDL, simulating the composition and size of newborn HDL. LCAT activity is critical to the maturation of newborn HDL granules into globular plasma HDL while maintaining the normal lipoprotein granule structure. Lecithin–cholesterol acyltransferase (LCAT) reverses cholesterol transport on the lung surface [63]. The amino acid sequence of LCAT is 50% of LPLA2, which is associated with high-density and low-density lipoprotein (HDL and LDL) particles in plasma. Moreover, it catalyzes the reverse transport of cholesterol from peripheral tissues to the liver [59]. The acyltransferase activity of LCAT esterifies free cholesterol on the discoid anterior b-HDL particles, maturing into spherical a-HDL. LCAT gene mutation is responsible for somatic diseases, such as familial LCAT deficiency (FLD) and fish-eye disease (FED) 10, due to complete loss of LCAT activity, resulting from the matrix loss due to HDL particles by LCAT activity.

LCAT is an enzyme that plays a crucial role in human plasma lipoprotein metabolism and is very important in maintaining cholesterol homeostasis and controlling cholesterol transport in blood circulation [60]. LCAT also performs some auxiliary reactions unrelated to cholesterol. These reactions involve the esterification of Lyso PCs to PCs [74]. Previous studies have revealed that trans-unsaturated fatty acids inhibit lecithin–cholesterol acyltransferase and alter its location specificity. Omega-3 polyunsaturated fatty acids cannot be synthesized by the human body. They are essential nutrients for the human body to synthesize various hormones and endogenous substances. The physiological functions of the body operate normally only by supplementing these oleic acids with food from the outside.

Our results indicate that H460 cells regulate enzyme and metabolite levels to resist when exposed to linolenic acid. Linolenic acid induced significant changes in LCAT, while the other three fatty acids induced no changes in LCAT. Therefore, H460 cells exposed to linolenic acid experience more cholesterol reverse transcription or Lyso PC esterification to PCs than those exposed to other fatty acids. Previous studies have indicated that trans fatty acids can reduce LCAT activity. However, there is no study on cis fatty acids. This study compared the different effects of four fatty acids on LCAT.

The experimental results of qPCR and WB of linolenic acid and linoleic acid groups are relatively consistent. The qPCR and WB results of the linolenic acid administration group were consistent with the metabonomics experimental results. This confirmed our conjecture concerning the effect of omega-3 fatty acids on LCAT expression. However, the results of the linoleic acid group were not consistent with the metabonomics results. Through the KEGG metabolic pathway map, in addition to LCAT, 22 enzymes, such as PLA2G4B and PLA2G4D, affect the conversion of PCs to Lyso PCs (Figure 10). Our experimental results indicate that the effect of linolenic acid while converting PCs to Lyso PCs may come from the other enzymes.

## 4. Materials and Methods

### 4.1. Chemicals and Reagents

HPLC-grade methanol, acetonitrile, and chloroform were obtained from Merck (Darmstadt, Germany). HPLC-grade ammonium formate and acetate were procured from Shanghai Aladdin Reagent Co., Ltd. (Shanghai, China). The energy metabolite standards were purchased from Shanghai Anpel Laboratory Technologies (Shanghai, China) Inc. (Shanghai, China). Phospholipid standards were obtained from Avanti Polar Lipids Inc. (Alabaster, AL, USA). Bile acid standards were bought from Sigma-Aldrich (St. Louis, MO, USA). Deionized water was produced using a Direct-Q water purification system (Millipore, El Paso, TX, USA). The target compound information is demonstrated in Appendix A.

### 4.2. Cultured Cells

H460 lung cancer cells were obtained from Bohui Biotechnology Co., Ltd. (Guangzhou, China). Using RPMI-1640, 10% FBS and 1% P/S were added as a complete medium. All H460 cell culture dishes were incubated at 5% CO_2_, 37 °C constant temperature, and saturated humidity and cultured using the completely prepared medium. Based on the cell growth rate and medium color, fresh medium was replaced in time. Before sample preparation, cells were treated with 500 μM of sodium acetate, butyrate, linoleic acid, or linolenic acid for 24 h.

### 4.3. CCK8 Experimental Procedure

After counting the cells, 100 microliters of 2000 cells per well were transferred into a 96-well plate. Simultaneously, a blank group was set, and a PBS buffer was added to the most marginal hole to decrease the evaporation effect. The cells were placed in a 37 °C, 5% CO_2_ incubator for 24 h. The adherence of the cells was observed, and the medium was aspirated from each well. Then, different concentrations of sodium acetate, sodium butyrate, linoleic acid, and linolenic acid were added to the experimental group. Following that, the fatty-acid-free medium was added to the blank group and placed in a 96-well plate at 37 °C. The medium was incubated for 24 h in a cell incubator with 5% CO_2_ air. After the administration, 10 μL of CCK-8 solution was directly added to each well. The culture plate was placed in a 37 °C 5% CO_2_ incubator for 1 h after adding CCK-8. The 96-well plate was taken out, a microplate reader was used to detect the OD value of each well at a wavelength of 450 nm, the processing data were analyzed, and the proliferation curve was drawn.

### 4.4. LC-MS/MS Sample Preparation

The cells grown in the log phase were taken, the medium was aspirated, and the cells were washed three times using PBS. We added 0.5 mL of pre-chilled methanol at 4 °C, used a cell scraper to scrape off the adherent cells, and transferred them to a 2 mL centrifuge tube. A specific concentration of internal isotopic standard was added to each sample standard. Then, 1 mm grinding beads were added to homogenize the sample using a homogenizer, at 300 Hz, for 60 s and three cycles. After that, 250 μL of water and chloroform were added and homogenized for three cycles. Then, it was centrifuged at 14,000 rpm for 10 min at 4 °C, and the supernatant was collected, spin-dried, and reconstituted using 400 μL of 50% acetonitrile/water solution. Samples were stored at −20 °C for LC-MS/MS analysis.

### 4.5. Untargeted Metabonomics

For HILIC untargeted metabonomics, the chromatographic separation was performed using a Vanquish UHPLC system (Thermofisher, Massachusetts, USA. with a binary pump, a vacuum degasser, an autosampler, and a column oven) in a SeQuant ZIC-HILIC column (100 mm × 2.1 mm i.d., 3.5 µm) (Merck, Germany) at 45 °C. Mobile phase A was water containing 50 mM of ammonium formate, and mobile phase B was acetonitrile. The linear gradient was 0 min, 90% B; 10 min, 50% B; 12 min, 90% B; 15 min, 90% B. The flow rate was 0.4 mL/min, and the injection volume was 1 μL.

For RP untargeted metabonomics, the chromatographic separation was performed using a Vanquish UHPLC system (ThermoFisher with a binary pump, a vacuum degasser, an autosampler, and a column oven) in an ACQUITY UPLC HSS T3 column (100 mm × 2.1 mm i.d., 1.7 µm) (Merck, Germany) at 45 °C. Mobile phase A was water containing 0.1% formic acid, and mobile phase B was acetonitrile with 0.1% formic acid. The linear gradient was 0 min, 1% B; 12 min, 90% B; 13 min, 90% B; 13.1 min, 1% B. The flow rate was 0.4 mL/min, and the injection volume was 1 μL. MRM analysis was performed with the Q Exactive plus mass spectrometer (ThermoFisher, Waltham, MA, USA) for untargeted metabonomics. The optimized MS parameters were HESI source in negative mode; scan mode: DDA mode, and one full scan followed by five MS/MS scans. The collision energy is NEC 15, 30, 45 to fragment the ions. Nitrogen (99.999%) was utilized as a collision-induced dissociation gas. Full scan range: 70 to 1000 amu; full scan resolution: 70000, AGC: 1e6, IT: 100ms; dd-MS/MS resolution: 17500, AGC: 5e5, IT: 50ms; spray voltage: 3.2 kV (positive mode) and 3.0 kV (negative mode); capillary temperature: 320 °C; S-lens RF level: 50 V. The data acquisition and processing were undertaken using the Analyst software 1.7 (AB SCIEX) and the MultiQuant software 3.0.3 (AB SCIEX).

### 4.6. Targeted Metabonomics

#### 4.6.1. Calibrators and Quality Control Samples

Standard stock solution: the energy metabolites and the bile acid standards were dissolved in methanol, and phospholipid standards were dissolved in chloroform to synthesize 1 mg/mL stock solutions, diluted using methanol to standard solutions.

Standard mixture: Appropriate amounts of standard stock solutions and deuterated internal standards were precisely obtained to prepare a standard mixture with a concentration of 10× (10 ng/μL). The mixed solvent gradient dilution was 3×, 1×, 0.7×, 0.5×, 0.3×, 0.1×, 0.07×, 0.05×, 0.03×, 0.01×, and 0.007× as the standard linear working solution.

#### 4.6.2. Method Validation

The accuracy and precision were determined for three days over one week (n = 6).

The stability profiles of the analytes were evaluated after storing at 4 °C and room temperature for three days and −20 °C for 20 days.

The current study used the internal standard to validate the extraction recovery and matrix effect. The MQC = 0.3× (n = 3) sample was selected for verification. The extraction recovery was the percentage of the peak area extracted after adding the internal standard to the peak area of the solution. The matrix effect was the ratio of the peak area of the internal standard added after the extraction and the internal standard.

#### 4.6.3. LC-MS/MS Parameters for Targeted Metabonomics

For the energy metabolites, chromatographic separation was performed using a UHPLC system (Waters Corp., Milford, MA) and a Waters BEH amide column (100 mm × 4.6 mm i.d., 3.5 μm) at 25 °C. Mobile phase A was 10 mM of ammonium acetate in 10:90 acetonitrile/water, with PH = 9, and mobile phase B was 10 mM of ammonium acetate in 90:10 acetonitrile/water, with PH = 9. The linear gradient was 0 min, 90% B; 2 min, 90% B; 12 min, 60% B; 15 min, 60% B; 15.1 min, 90% B; 18 min, 90% B. The flow rate was 0.4 mL/min, and the injection volume was 10 μL.

For phospholipids and bile acids, chromatographic separation was performed using a UHPLC system (Waters Corp., Milford, MA, USA) and a Waters BEH C18 column (100 mm × 4.6 mm i.d., 3.5 μm) at 50 °C. Mobile phase A involved 5 mM of ammonium formate in 5:95 acetonitrile/water, and mobile phase B was acetonitrile. The linear gradient used was 0 min, 27% B; 0.1 min, 27% B; 1.5 min, 32% B; 4.5 min, 35% B; 7 min, 35% B; 10 min, 34% B; 10.01 min, 95% B; 13 min, 100% B; 16 min, 100% B; 16.01 min, 27% B; and 20 min, 27% B. The flow rate was 0.3 mL/min, and the injection volume was 10 μL.

MRM analysis was performed for energy metabolites using the QTRAP 4000, a hybrid triple quadrupole/linear ion trap (AB SCIEX, Concord, ON, Canada), with the following optimized ESI- MS parameters: collision gas, medium, curtain gas, 20 psi; ion source gas 1, 50 psi; ion source gas 2, 50 psi; ion spray voltage, 5000 V and source temperature, 500 °C. Additionally, the data acquisition and processing were conducted using the Analyst software 1.7 (AB SCIEX) and the MultiQuant software 3.0.3 (AB SCIEX).

MRM analysis was performed with QTRAP 4000, a hybrid triple quadrupole/linear ion trap (AB SCIEX, Concord, ON, Canada) for phospholipids and bile acids.

The optimized ESI ± MS parameters were collision gas, medium, curtain gas, 15 psi; ion source gas 1, 50 psi; ion source gas 2, 50 psi; ion spray voltage, ±4500 V and source temperature, at 500 °C. The data acquisition and processing were undertaken using the Analyst software 1.7 (AB SCIEX) and the MultiQuant software 3.0.3 (AB SCIEX).

Detailed mass spectrum parameters of all the target compounds are shown in Appendix A.

### 4.7. PCR Experimental Procedure

After RNA extraction, reverse transcription experiments were conducted to convert RNA to cDNA. Then, PCR amplification experiments were performed. The sequence of primers is depicted in Appendix A.

### 4.8. WB Experimental Procedure

After protein extraction, the protein concentration was determined with the BCA method. After adding the loading buffer, the protein was denatured by boiling. Subsequent SDS-PAGE electrophoresis was performed until the rainbow markers were entirely separated. The electrophoresis gel was cut to a suitable size and transferred to the membrane. After transfer, the membrane was blocked using 5% BSA for 1 h. The blocked membrane was removed, the diluted primary antibody solution was added, and it was kept in a refrigerator at 4 °C overnight. After washing three times using TBST, the diluted secondary antibody solution was added and incubated for 1 h. Then, we performed development after three washes with TBST.

### 4.9. Statistical Analysis of Data

The collected raw data were processed using Analyst 1.4.2 and Masslynx 4.1, and the target concentration was obtained using the standard curve.

Statistical analyses were performed with Simca-P (Sartorius Company, Goettingen, Germany) and GraphPad Prism (GraphPad Software Inc., San Diego, CA, USA). The overall distribution of the unsupervised PCA samples and the supervised OPLS-DA methods are utilized to observe the differences between the groups. The response sequencing test and cyclic interactive verification methods are simultaneously used to prevent the model from overfitting and verify the stability and prediction accuracy. In multi-dimensional statistics, VIP > 1 is considered a variance variable. We process the single dimension data statistically, with Student’s *t*-tests and multiple changes, and finally select metabolites having ANOVA *p*-value < 0.05, fold change > 1.5, and maximum CV < 30% as the difference variable for identification. MetaboAnalyst 4.0 heatmap, correlation analysis, and other means were used to search for potential lung cancer biomarkers with high specificity and sensitivity to understand the correlation of differential metabolites. * *p* < 0.05 and ** *p* < 0.01 were considered statistically significant in all the experiments.

## 5. Conclusions

A metabolomics approach was used to identify the metabolic perturbations in H460 cells induced by sodium acetate, sodium butyrate, linoleic acid, and linolenic acid. The results obtained in this study showed that exposure to four fatty acids would cause significant changes in the metabolic profile of H460 cells compared with the control group. Changes in the metabolic spectrum involve multiple metabolic pathways, such as the citrate cycle (TCA cycle), gluconeogenesis, and glycerophospholipid metabolism. In addition, the results of the metabonomic analysis were also verified, indicating that the expression of LCAT changed after administration. Additionally, the administration of four fatty acids caused unique changes in H460 cells. This showed that although they had several core similarities, each compound had different effects on cells. For instance, sodium butyrate administration leads to a more potent disturbance of the cell metabolic spectrum than sodium acetate. However, sodium butyrate had a weaker effect on LCAT expression than linolenic acid. Our results demonstrate that metabonomics is a powerful tool for assessing health status and provides data for the overall health risk assessment of fatty acid intake.

## Figures and Tables

**Figure 1 molecules-28-02357-f001:**
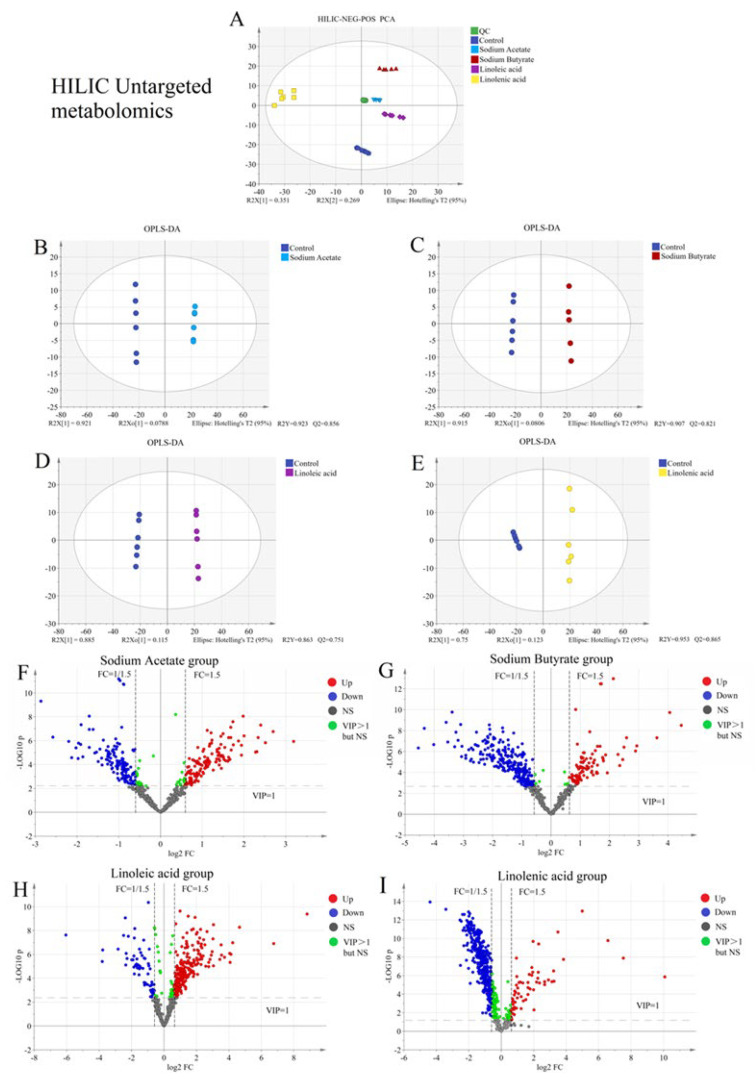
Screening differential metabolites between the control group and four fatty acid groups based on HILIC untargeted metabonomics: (**A**) PCA plots; (**B**–**E**) OPLS-DA plots; (**F**–**I**) volcano plots of sodium acetate group, sodium butyrate group, linoleic acid group, and linolenic acid group compared with the control group.

**Figure 2 molecules-28-02357-f002:**
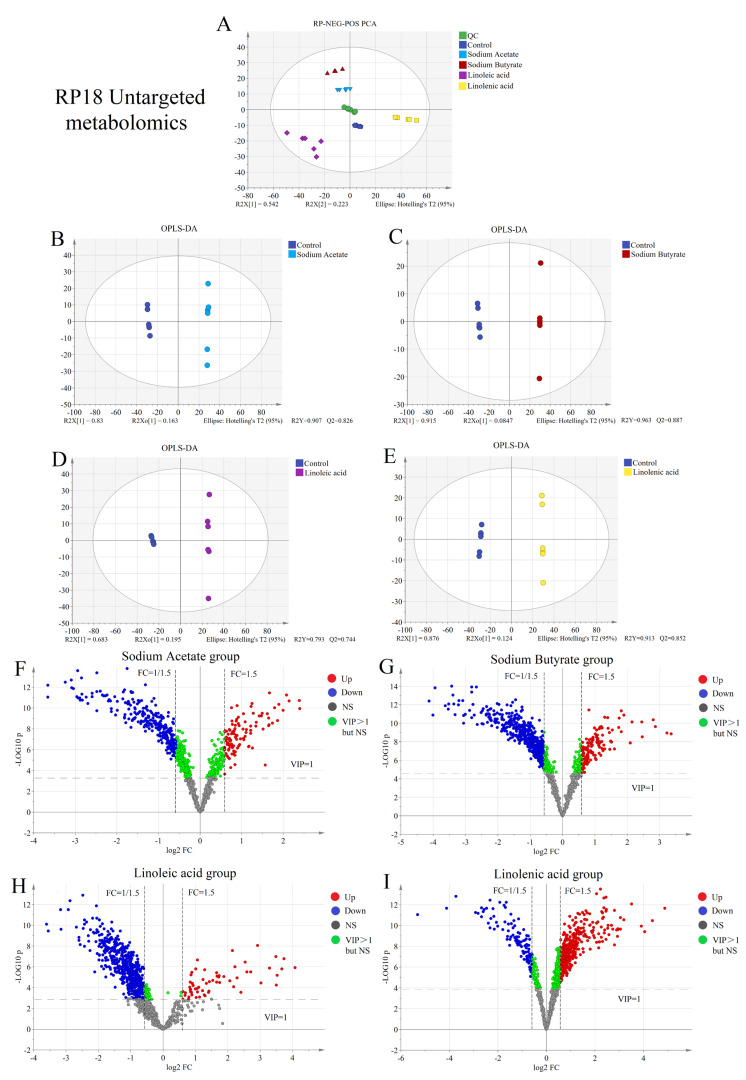
Screening differential metabolites between the control group and four fatty acid groups based on RP18 untargeted metabonomics: (**A**) PCA plots; (**B**–**E**) OPLS-DA plots; (**F**–**I**) volcano plots of sodium acetate group, sodium butyrate group, linoleic acid group, and linolenic acid group compared with the control group.

**Figure 3 molecules-28-02357-f003:**
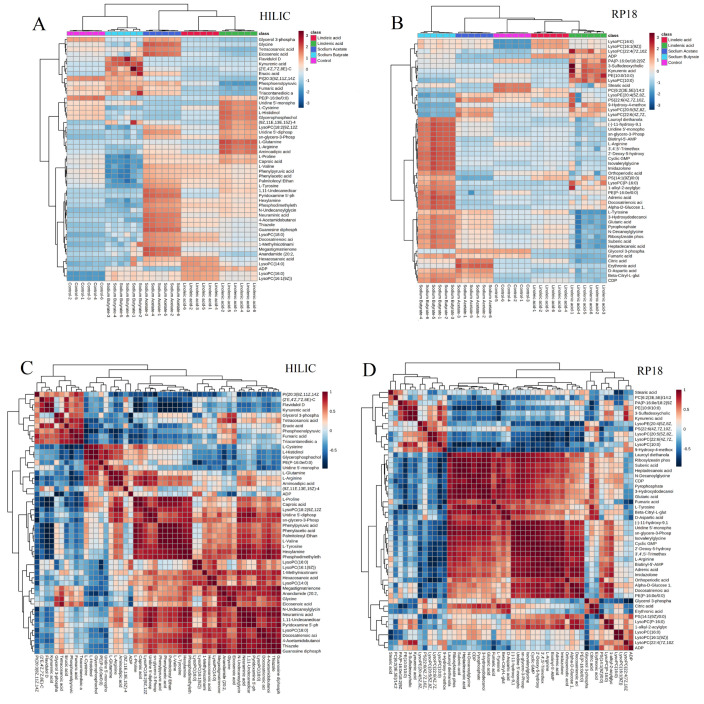
Heatmaps and correlation analysis depending on untargeted metabonomics: (**A**) heatmap based on HILIC untargeted metabonomics, (**B**) heatmap depending on RP-18 untargeted metabonomics, (**C**) correlation analysis based on HILIC untargeted metabonomics, and (**D**) correlation analysis depending on RP-18 untargeted metabonomics.

**Figure 4 molecules-28-02357-f004:**
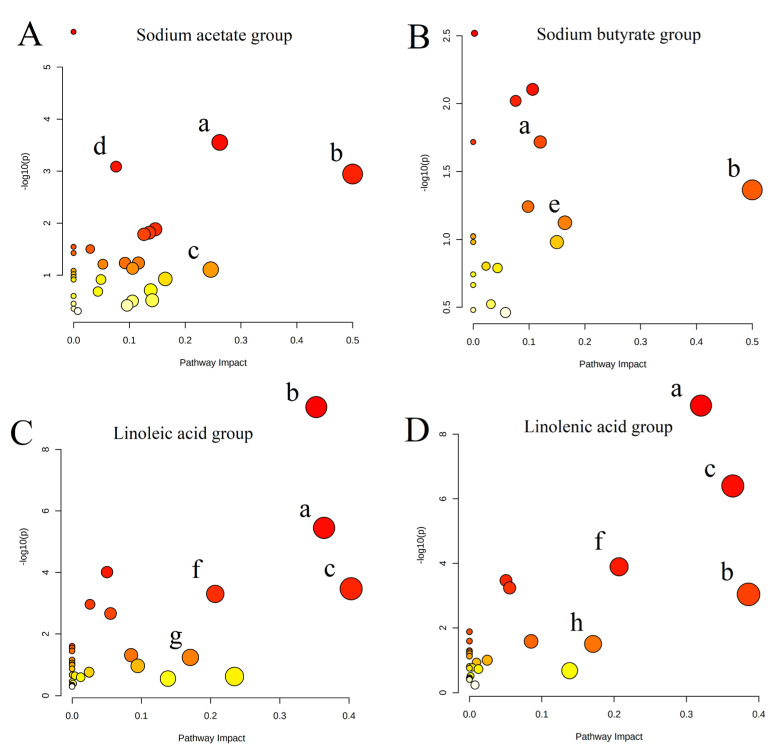
Global metabolic pathways are affected by four fatty acid groups: (**A**) sodium acetate, (**B**) sodium butyrate, (**C**) linoleic acid, and (**D**) linolenic acid. The following abbreviations were utilized: a—glycerophospholipid metabolism; b—citrate cycle (TCA cycle); c—gluconeogenesis; d—Aminoacyl-tRNA biosynthesis; e—phenylalanine metabolism; f—arginine biosynthesis; g—pyrimidine metabolism; h—pyruvate metabolism.

**Figure 5 molecules-28-02357-f005:**
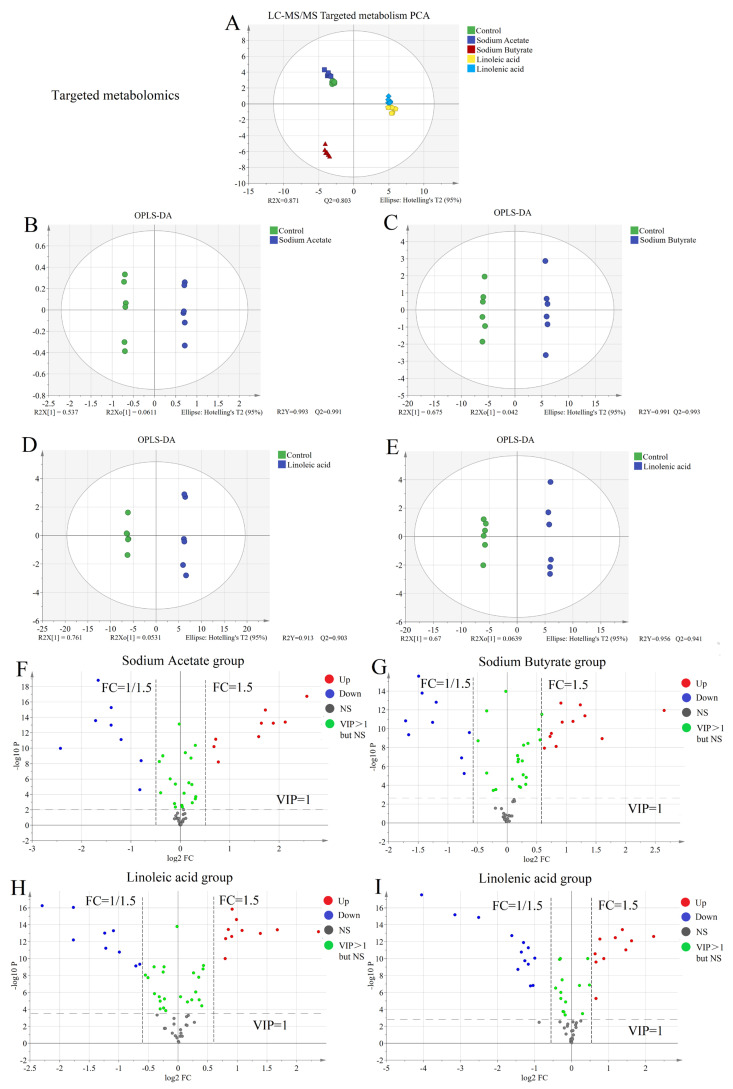
Screening differential metabolites between the control and the four fatty acid groups based on targeted metabonomics: (**A**) PCA plots; (**B**–**E**) OPLS-DA plots; (**F**–**I**) volcano plots of sodium acetate group, sodium butyrate group, linoleic acid group, and linolenic acid group compared with the control group.

**Figure 6 molecules-28-02357-f006:**
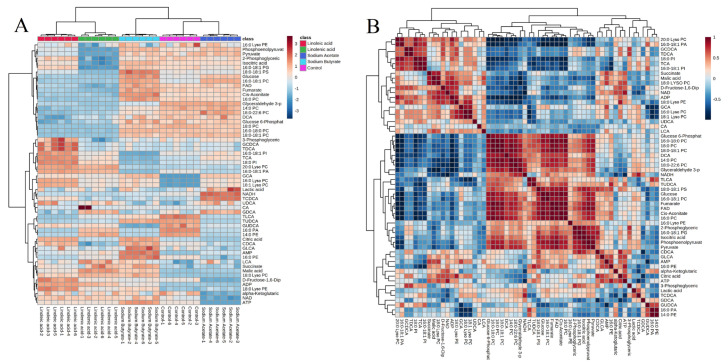
Heatmap (**A**) and correlation analysis (**B**) based on targeted metabonomics.

**Figure 7 molecules-28-02357-f007:**
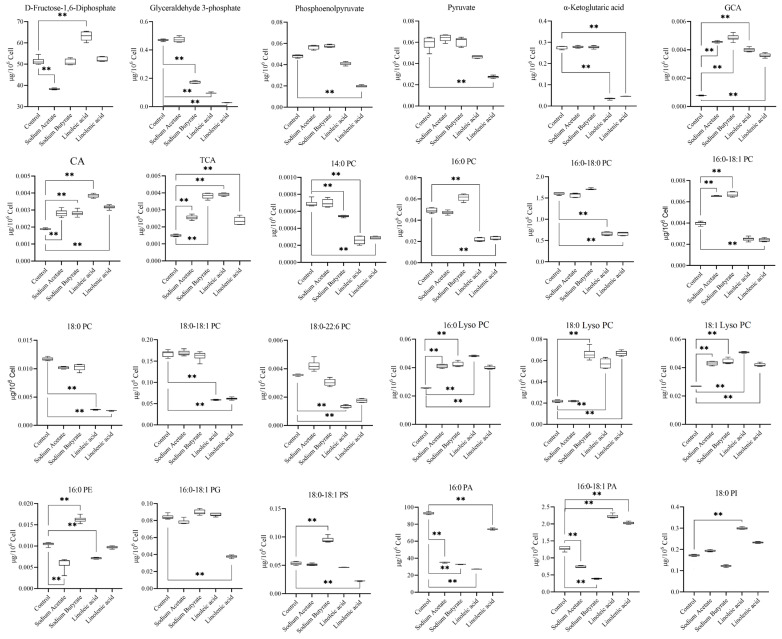
Box plot representing the differential metabolites in targeted metabonomics. **: *p* value < 0.01.

**Figure 8 molecules-28-02357-f008:**
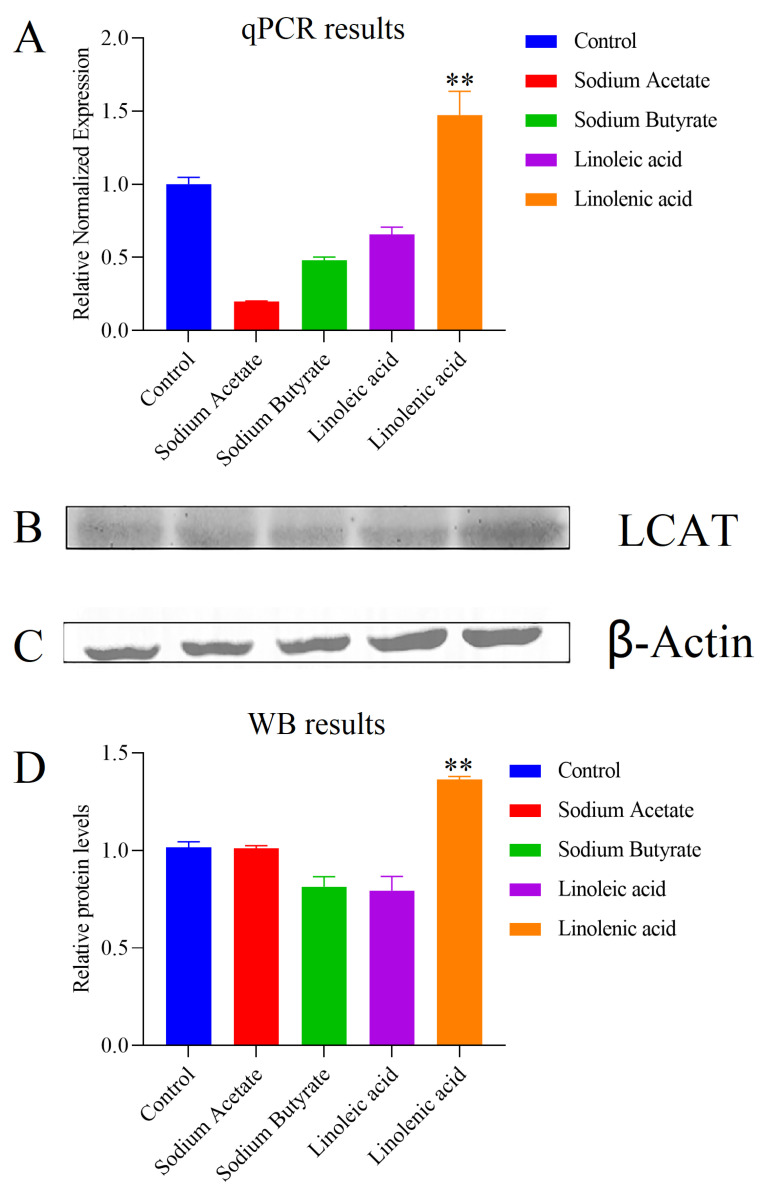
qPCR and WB results based on targeted metabonomics: (**A**) Effects on mRNA levels of LCAT in H460 cells treated with sodium acetate, sodium butyrate, linoleic acid, and linolenic acid for 24 h, n = 3. (**B**) WB results of LCAT; from left to right are the control group, sodium acetate group, sodium butyrate group, linoleic acid group, and linolenic acid group. (**C**) WB results of β-Actin; 17rom left to right are the control group, sodium acetate group, sodium butyrate group, linoleic acid group, and linolenic acid group. (**D**) Effects on protein content in H460 cells treated with sodium acetate, sodium butyrate, linoleic acid, and linolenic acid for 24 h; n = 6; **: *p* value < 0.01.

**Figure 9 molecules-28-02357-f009:**
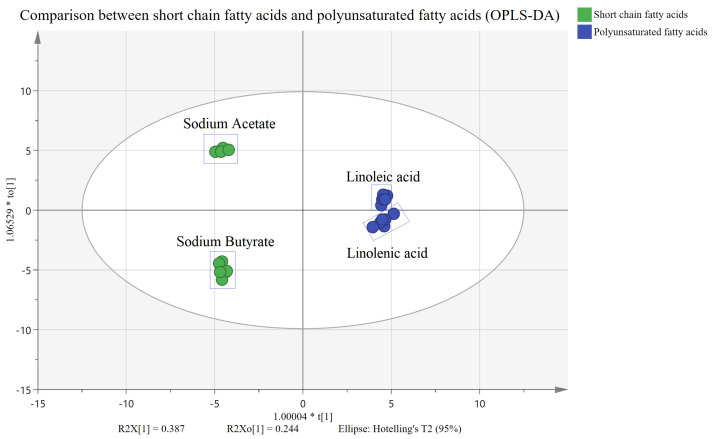
OPLS-DA plots between short-chain and polyunsaturated fatty acids.

**Figure 10 molecules-28-02357-f010:**
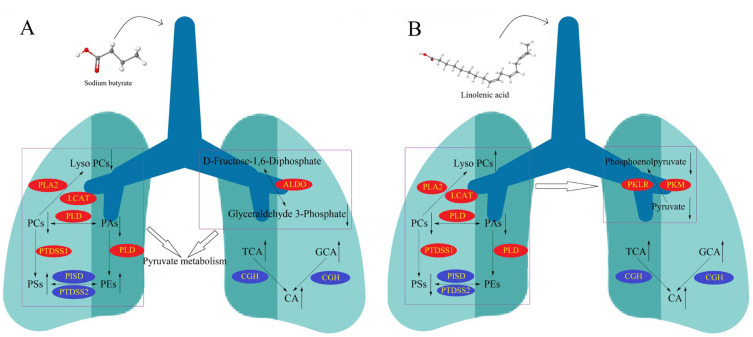
(**A**) Diagram showing sodium-butyrate-induced lung cancer molecular mechanism. (**B**) Diagram showing linolenic-acid-induced lung cancer molecular mechanism; Blue ellipse: down-regulated. Red ellipse: up-regulated.

**Table 1 molecules-28-02357-t001:** Linear range, LOD, LOQ, and precision data of the target compound.

Compound	Linear Range/μg·mL^−1^	LOD/μg·mL^−1^	LOQ/μg·mL^−1^	LQC/0.03X			MQC/0.3X			HQC/1X		
				RE%	CV%	CV%	RE%	CV%	CV%	RE%	CV%	CV%
					Intra (n = 5)	Inter (n = 5)		Intra (n = 5)	Inter (n = 5)		Intra (n = 5)	Inter (n = 5)
Pyruvate	0.033–2.333	0.011	0.037	6.46	12.46	9.59	12.00	9.52	12.41	6.42	2.24	3.82
Lactic acid	0.023–2.333	0.008	0.027	6.91	9.82	9.38	8.94	8.36	6.70	8.80	3.95	4.45
Fumarate	0.033–2.333	0.011	0.037	11.22	6.55	13.95	13.78	8.17	7.31	14.28	6.59	1.73
Succinate	0.033–2.333	0.011	0.037	14.76	7.64	8.53	11.95	7.37	7.74	8.31	2.32	7.94
Malic acid	0.023–2.333	0.008	0.027	14.06	7.79	13.19	6.71	13.43	13.58	9.60	2.31	3.08
α-Ketoglutaric acid	0.233–10	0.766	2.553	5.91	6.00	11.83	5.54	8.73	10.65	14.92	9.71	8.87
Phosphoenolpyruvate	0.033–2.333	0.011	0.037	12.21	9.79	7.72	10.75	7.02	9.76	9.63	0.64	6.71
Glyceraldehyde 3-phosphate	0.033–2.333	0.011	0.037	7.72	10.26	13.42	12.60	13.40	14.86	13.57	2.88	1.53
Cis-Aconitate	0.003–2.333	0.001	0.003	10.79	14.12	6.53	10.52	10.75	11.96	6.44	2.51	1.55
Glucose	0.033–2.333	0.011	0.037	12.98	8.92	6.75	10.44	6.25	10.88	13.61	8.07	0.41
2-Phosphoglyceric acid	0.233–10	0.766	2.553	11.96	10.18	7.51	11.66	14.33	14.79	14.00	4.78	8.91
3-Phosphoglyceric acid	0.033–2.333	0.011	0.037	10.67	10.50	13.10	12.60	12.15	9.95	14.23	2.72	0.63
Citric acid	0.033–2.333	0.011	0.037	9.20	13.83	5.36	8.01	6.23	8.57	10.36	7.97	0.20
Isocitric acid	0.033–2.333	0.011	0.037	11.03	7.48	14.66	8.28	6.57	8.85	13.81	8.85	0.94
Glucose 6-Phosphate	3.333–233.333	1.111	3.703	12.41	5.36	8.14	5.90	10.87	13.68	9.89	5.94	3.90
D-Fructose-1,6-Diphosphate	3.333–233.333	1.111	3.703	14.18	5.45	6.36	11.59	13.39	6.30	12.45	7.83	2.48
AMP	0.033–2.333	0.011	0.037	7.78	8.58	12.01	13.97	13.69	13.60	6.05	5.77	0.96
ADP	0.233–23.333	0.766	2.553	13.92	10.69	5.54	9.74	7.41	6.70	11.02	7.82	8.63
ATP	0.233–23.333	0.766	2.553	14.79	13.98	11.47	5.89	8.70	6.63	9.78	0.40	5.44
NAD	0.01–2.333	0.003	0.010	8.94	14.50	12.18	10.11	8.16	11.40	6.01	3.90	7.53
NADH	0.233–23.333	0.766	2.553	13.19	8.10	14.24	11.14	13.03	9.51	7.12	5.81	7.10
FAD	0.01–2.333	0.003	0.010	7.02	11.39	6.82	10.75	14.54	5.75	11.37	3.57	5.28
Acetyl CoA	1–33.333	0.333	1.110	10.91	5.47	13.19	12.80	7.78	14.88	14.30	4.76	2.22
TCA	0.005–0.35	0.005	0.017	2.40	3.29	2.84	3.18	1.37	0.67	3.69	2.58	2.43
GCA	0.005–0.35	0.005	0.017	4.04	3.26	4.33	4.77	3.69	1.67	2.34	3.79	1.13
CA	0.005–0.35	0.005	0.017	5.46	3.37	0.64	5.20	4.53	2.91	3.08	3.55	1.38
TCDCA	0.005–0.35	0.005	0.017	3.02	2.66	4.54	4.64	2.28	1.45	3.78	5.07	5.33
GCDCA	0.005–0.35	0.005	0.017	1.79	4.75	2.60	1.61	1.63	0.56	0.86	0.64	1.04
TDCA	0.005–0.35	0.005	0.017	1.15	3.43	2.79	1.11	1.07	3.60	4.68	1.01	2.81
CDCA	0.005–0.35	0.005	0.017	3.85	1.06	2.54	3.39	4.94	3.77	2.50	3.58	1.50
GDCA	0.005–0.35	0.005	0.017	1.99	1.99	1.27	5.33	5.40	0.99	5.26	4.20	3.35
DCA	0.005–0.35	0.005	0.017	4.77	2.47	1.30	2.05	0.59	1.73	3.73	3.07	1.47
TLCA	0.005–0.35	0.005	0.017	4.69	2.03	4.88	3.69	1.84	0.95	5.17	0.83	5.49
GLCA	0.005–0.35	0.005	0.017	5.36	5.38	2.94	3.77	3.85	3.30	3.02	4.27	5.47
LCA	0.005–0.35	0.005	0.017	3.75	0.51	1.12	3.00	4.22	2.50	5.48	2.52	5.06
TUDCA	0.005–0.35	0.005	0.017	1.49	4.08	3.85	0.73	4.01	4.04	4.26	4.06	4.85
GUDCA	0.005–0.35	0.005	0.017	0.99	3.65	4.45	0.83	3.61	1.53	5.18	1.29	1.37
UDCA	0.005–0.35	0.005	0.017	1.64	2.68	4.11	1.06	1.15	1.06	5.05	3.43	3.77
14:0 PC	0.03–0.7	0.03	0.100	4.79	3.27	0.51	1.00	1.10	1.40	4.87	2.01	3.44
16:0 PC	0.01–0.7	0.01	0.033	0.79	1.05	0.59	0.97	5.04	5.18	2.67	5.33	1.05
16:0–18:0 PC	0.01–0.7	0.01	0.033	2.04	3.43	2.67	4.19	2.79	4.10	3.63	1.74	1.16
16:0–18:1 PC	0.01–0.7	0.01	0.033	2.80	5.34	0.69	0.86	2.64	0.65	4.80	1.95	5.03
16:0–22:4 PC	0.01–0.7	0.01	0.033	3.05	2.03	0.83	0.65	4.62	3.32	4.73	2.65	5.46
18:0 PC	0.01–0.7	0.01	0.033	3.16	4.16	1.67	2.44	1.19	5.49	1.53	2.97	2.68
18:0–18:1 PC	0.01–0.7	0.01	0.033	1.28	0.73	2.46	1.55	1.66	4.53	0.62	4.82	0.64
18:0–22:6 PC	0.01–0.7	0.01	0.033	0.90	4.53	0.62	1.93	4.28	3.07	5.26	4.39	2.50
20:0 PC	0.01–0.7	0.01	0.033	3.34	2.35	3.73	4.81	2.34	2.81	3.79	1.75	2.95
16:0 Lyso PC	0.01–0.7	0.01	0.033	1.47	2.12	0.98	1.03	2.77	4.30	0.70	1.97	2.85
18:1 Lyso PC	0.01–0.7	0.01	0.033	5.44	2.38	1.03	5.00	3.82	1.93	1.65	4.12	2.59
18:0 Lyso PC	0.03–0.7	0.03	0.100	1.70	4.89	5.33	5.11	1.42	2.17	0.76	3.82	4.23
20:0 Lyso PC	0.01–0.7	0.01	0.033	2.85	3.71	2.24	2.12	4.26	3.73	2.73	1.89	1.53
16:0 PA	0.006–0.42	0.006	0.020	2.62	3.07	2.40	1.61	3.92	4.05	1.49	2.56	3.14
16:0–18:1 PA	0.006–0.42	0.006	0.020	1.83	4.01	3.78	2.28	4.24	3.62	1.31	2.04	2.30
16:0 Lyso PA	0.006–0.42	0.006	0.020	1.34	2.99	4.01	1.54	3.29	3.24	4.72	4.65	1.82
16:0 PS	0.006–0.42	0.006	0.020	2.56	4.85	1.55	3.62	3.29	1.22	1.79	4.66	1.53
18:0–18:1 PS	0.006–0.42	0.006	0.020	3.97	3.05	2.16	3.09	4.92	1.31	2.17	3.29	1.53
18:0 Lyso PS	0.006–0.42	0.006	0.020	1.44	3.76	4.75	3.23	1.17	4.45	4.05	2.29	3.25
16:0 PI	0.02–0.14	0.02	0.067	1.02	4.20	4.59	4.72	4.32	1.44	2.54	3.54	1.93
18:0 PI	0.02–0.14	0.02	0.067	3.52	4.26	3.32	4.37	1.65	4.23	1.92	3.65	3.74
16:0–18:1 PI	0.014–0.14	0.014	0.047	1.52	4.66	1.69	3.99	1.73	4.10	4.72	3.02	3.25
18:0–20:4 PI	0.006–0.14	0.006	0.020	1.57	1.31	4.55	1.37	4.53	1.24	3.70	2.81	1.20
18:0 Lyso PI	0.006–0.14	0.006	0.020	3.11	2.49	2.05	3.23	2.53	2.85	2.21	2.11	2.21
14:0 PG	0.025–0.35	0.025	0.083	1.57	2.22	3.29	3.80	2.46	4.67	2.07	4.84	4.34
16:0–18:1 PG	0.005–0.35	0.005	0.017	3.50	2.59	1.85	3.52	2.64	3.78	3.75	3.79	2.15
18:0 PG	0.005–0.35	0.005	0.017	2.55	1.10	4.61	4.81	2.24	1.27	2.43	1.29	4.88
14:0 PE	0.005–0.35	0.005	0.017	1.85	1.02	1.19	2.53	1.52	2.77	1.31	4.38	4.68
16:0 PE	0.005–0.35	0.005	0.017	4.69	2.95	4.99	1.13	4.04	3.96	3.76	2.68	4.27
18:0 PE	0.005–0.35	0.005	0.017	3.51	2.85	1.03	2.24	3.22	1.29	1.15	4.49	4.64
16:0 Lyso PE	0.005–0.35	0.005	0.017	3.52	2.86	3.25	1.13	2.19	1.53	2.34	3.61	2.22
18:0 Lyso PE	0.005–0.35	0.005	0.017	3.60	1.14	3.51	1.63	3.74	4.70	4.13	4.71	2.20

**Table 2 molecules-28-02357-t002:** Recovery and matrix effect data of the target compound.

IS	MQC (n = 6)	
	Recovery Rate (% ± SD)	Matrix Effect (% ± SD)
^13^C-GCA	95.2 ± 2.9	101.2 ± 2.1
17:0 Lyso PC	85.2 ± 4.5	93.4 ± 5.8
17:0 Lyso PA	81.9 ± 3.6	89.4 ± 6.9
17:1 Lyso PS	84.4 ± 8.6	95.8 ± 8.1
17:1 Lyso PI	88.4 ± 1.9	102.7 ± 2.2
15:0 PG	85.6 ± 3.3	87.6 ± 2.1
17:1 Lyso PE	94.6 ± 4.5	93.5 ± 3.4
^13^C-AMP	86.6 ± 2.9	94.8 ± 2.8
d_4_-Succinic acid	81.9 ± 5.2	89.6 ± 4.3

**Table 3 molecules-28-02357-t003:** Stability data of the target compound.

IS	Cell (n = 6)	
	Short-Term Stability (% ± SD)	Long-Term Stability (% ± SD)
^13^C-GCA	98.2 ± 2.2	99.1 ± 3.8
17:0 Lyso PC	88.9 ± 10.9	100.5 ± 9.8
17:0 Lyso PA	86.2 ± 8.2	100.4 ± 7.1
17:1 Lyso PS	90.8 ± 3.2	91.1 ± 7.9
17:1 Lyso PI	88.9 ± 3.2	96.4 ± 1.9
15:0 PG	83.6 ± 9.8	91.6 ± 1.7
17:1 Lyso PE	95.7 ± 0.9	91.1 ± 0.3
^13^C-AMP	88.7 ± 5.6	92.7 ± 2.1
d_4_-Succinic acid	91.8 ± 1.4	85.9 ± 0.8
	Short-term stability (% ± SD): 4 °C 24 h	
	Long-term stability (% ± SD): −20 °C 2 months	

## Data Availability

All relevant data are included in the article.

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
