# Peer review of "Effect of Short-Chain Fatty Acids and Polyunsaturated Fatty Acids on Metabolites in H460 Lung Cancer Cells"

_molecules, 2023, doi:10.3390/molecules28052357_

Round 1
Reviewer 1 Report
Major Comments:
1) None of the figures are self-explanatory neither are arranged properly for the ease of the reader. Also, please provide the detailed legends for all the figures.
2) As Fig 1A , Fig 2A-D, Fig3A,D explains the analysis depending on HILIC untargeted metabonomic, whereas Fig 1B, Fig 2E-H, Fig 3C,D shows analysis based on RP-18, can they be put together?
2) In Fig 8, authors need to provide the experimental number (n) as well as it will be helpful if they provide individuals values with box graph in order to understand the variation among the sample.
2) Fig 8A showed increased LCAT expression in H460 lung cancer cell line, however, authors state that the levels remain unchanged and thus were validated with WB experiment. Is that a technical error?
3) For WB experiments, authors need to provide the LCAT and Actin (control) expression as one/ uncut blot and Fig 8B-C can't be separated. Also, provide the statistical analysis for the Fig 8D.
Minor Comments:
1) The resolution of the figure 3 can be improved to be more readable
2) Line 316: Fig 8 is mislabelled as Fig 1
2) Figure 2, the volcano plots need the detailed information
Author Response
Response to Reviewer 1 Comments
Dear editor and reviewer,
First of all, we are very grateful for giving us the opportunity to revise our manuscript (molecules-2171777) entitled “Effect of short-chain fatty acids and polyunsaturated fatty acids on metabolites in H460 lung cancer cells”. We appreciate the time and effort that you and the reviewers dedicated to providing insightful and valuable feedback on our manuscript. We have thought carefully about the reviewers’ suggestions and concerns, and the manuscript has been revised accordingly. Our reply is marked in red while the changes made are highlighted in yellow in the manuscript or supporting information and are presented in red in the following response. Please, also note that extra figures and tables containing unpublished data and/or adapted from relevant references were added in this reply only for review to assist in addressing the reviewers’ concerns; these figures and tables are named “figure E” and “table E” followed by a serial number in the order of their appearance and they are not included in the manuscript nor in the supporting information.
Point 1: None of the figures are self-explanatory neither are arranged properly for the ease of the reader. Also, please provide the detailed legends for all the figures.
Response 1: Thank you for reviewer’s comments. We have re-arranged some of the images in Figures 1 and 2 for improved readability and added more detailed labels and annotations to make the figures self-explanatory. The order of the images has also been adjusted according to the experimental sequence for better organization. The figure legends have been modified accordingly and the manuscript has been updated. Please find below the revised figure legends.
Figure 1. Screening differential metabolites between the control group and four fatty acids group based on HILIC untargeted metabonomics: (A) PCA plots, (B-E) OPLS-DA plots, (F-I) volcano plots of sodium acetate group, sodium butyrate group, linoleic acid group and linolenic acid group compared with the control group.
Figure 2. Screening differential metabolites between the control group and four fatty acids group based on RP18 untargeted metabonomics: (A) PCA plots, (B-E) OPLS-DA plots, (F-I) volcano plots of sodium acetate group, sodium butyrate group, linoleic acid group and linolenic acid group compared with the control group.
Figure 3. Heatmaps and correlation analysis based on untargeted metabonomics: (A) Heatmap based on HILIC untargeted metabonomics, (B) Heatmap based on RP-18 untargeted metabonomics, (C) Correlation analysis based on HILIC untargeted metabonomics, (D) Correlation analysis based on RP-18 untargeted metabonomics.
Figure 4. Global metabolic pathways affected by the four fatty acids: (A) sodium acetate group, (B) sodium butyrate group, (C) linoleic acid group, and (D) linolenic acid group. The following abbreviations were used: (a) glycerophospholipid metabolism, (b) Citrate cycle (TCA cycle), (c) Gluconeogenesis, (d) Aminoacyl-tRNA biosynthesis, (e) Phenylalanine metabolism, (f) Arginine biosynthesis, (g) Pyrimidine metabolism and (h) Pyruvate metabolism.
Figure 5. Screening differential metabolites between the control group and four fatty acids group based on targeted metabonomics: (A) PCA plots, (B-E) OPLS-DA plots, (F-I) volcano plots of sodium acetate group, sodium butyrate group, linoleic acid group and linolenic acid group compared with the control group.
Figure 6. Heatmap (A) and correlation analysis(B) based on targeted metabonomics:
Figure 7. Box plot of differential metabolites in targeted metabonomics.
Figure 8. qPCR and WB results based on targeted metabonomics: (A) Effects on mRNA levels of LCAT in H460 cells treated with sodium acetate, sodium butyrate, linoleic acid and linolenic acid for 24 h, n=3, (B) WB results of LCAT, From left to right are control group, sodium acetate group, sodium butyrate group, linoleic acid group, linolenic acid group, (C) WB results of β-Actin, From left to right are control group, sodium acetate group, sodium butyrate group, linoleic acid group, linolenic acid group, (D) Effects on protein content in H460 cells treated with sodium acetate, sodium butyrate, linoleic acid and linolenic acid for 24 h; n=6.
Figure 9. OPLS-DA plots between short-chain fatty acids and polyunsaturated fatty acids.
Figure 10. (A) Diagram of sodium butyrate-induced lung cancer molecular mechanism, (B) Diagram of linolenic acid-induced lung cancer molecular mechanism.
Point 2: As Fig 1A , Fig 2A-D, Fig3A,D explains the analysis depending on HILIC untargeted metabonomic, whereas Fig 1B, Fig 2E-H, Fig 3C,D shows analysis based on RP-18, can they be put together?
Response 2: Thank you for the reviewers' comments. According to your suggestion, we integrated the PCA, OPLS-DA, and volcano plots for HILIC untargeted metabolomics into Figure 1, and for RP18 untargeted metabolomics into Figure 2. As the heatmap and correlation heatmap in Figure 3 have relatively large image sizes, their integration into Figures 1 and 2 would compromise the overall visual presentation. Therefore, with respect to Figure 3, we marked the corresponding HILIC and RP18 untargeted metabolomics identifiers on the figure.
Point 3: In Fig 8, authors need to provide the experimental number (n) as well as it will be helpful if they provide individuals values with box graph in order to understand the variation among the sample.
Response 3: Thank you for reviewer’s comments. For the Western blot experiments, a sample size of n=3 was used. In the qPCR experiments, a sample size of n=6 was used. Also, we have added the 'n' values for Figure 8A and Figure 8D to the legend of Figure 8.
Please see Table E1 for WB data, and Table E2 for qPCR data
Table E1. Statistical analysis of WB
Control |
Sodium Acetate |
Sodium Butyrate |
Linoleic acid |
Linolenic acid |
0.98759 |
0.994867 |
0.767771 |
0.712307 |
1.312595 |
1.01468 |
1.021645 |
0.869456 |
0.84564 |
1.345464 |
1.04546 |
1.014564 |
0.804456 |
0.825341 |
1.325466 |
Point 4: Fig 8A showed increased LCAT expression in H460 lung cancer cell line, however, authors state that the levels remain unchanged and thus were validated with WB experiment. Is that a technical error?
Response 4: Thank you for the reviewers' comments. This is our clerical error. We have modified the first sentence of 2.5.2, The correct sentence is” The qPCR results showed that the four fatty acids affect the gene transcription level of LCAT in varying degrees.”
We tested whether four fatty acids altered the gene expression of LCAT (Figure 8A). H460 lung cancer cells were exposed to 500 μM sodium acetate, butyrate, linoleic acid, and linolenic acid. As shown in Figure 8A, LCAT expressions were differently affected by exposure to the four fatty acids. The expression level of LCAT was up-regulated during linolenic acid treatment while downregulated when treated with sodium acetate, butyrate, and linoleic acid. These results suggested that fatty acids significantly and differently affected H460 lung cancer cells. To further study the effect of four fatty acids on LCAT expression, we conducted a WB experiment. Western blot results (Figure 8B-D) revealed that linolenic acid induced the strongest LCAT expression. The other three fatty acids generated no significant changes in LCAT expression.
Point 5: For WB experiments, authors need to provide the LCAT and Actin (control) expression as one/ uncut blot and Fig 8B-C can't be separated. Also, provide the statistical analysis for the Fig 8D.
Response 5: Thank you for the reviewers' comments. The uncut blots for LCAT and ACTIN are presented in Figure E1 and Figure E2. The statistical analysis for Figure 8D is provided in Table E2.
Figure E1. Uncut blot result of ACTIN.
Figure E2. Uncut blot result of LCAT.
Table E2. Statistical analysis of qPCR
Target |
Sample |
Control |
Expression |
Expression SEM |
Corrected Expression SEM |
Mean Cq |
Cq SEM |
GAPDH |
Sodium Butyrate |
- |
N/A |
N/A |
N/A |
16.86 |
0.05847 |
LCAT |
Sodium Butyrate |
- |
0.48020 |
0.0223 |
0.0223 |
28.28 |
0.0328 |
GAPDH |
Sodium Acetate |
- |
N/A |
N/A |
N/A |
16.79 |
0.00707 |
LCAT |
Sodium Acetate |
- |
0.19951 |
0.0025 |
0.0025 |
29.47 |
0.01668 |
GAPDH |
Linolenic acid |
- |
N/A |
N/A |
N/A |
18.16 |
0.01419 |
LCAT |
Linolenic acid |
- |
1.38655 |
0.076215 |
0.076215 |
28.05 |
0.03406 |
GAPDH |
Linoleic acid |
- |
N/A |
N/A |
N/A |
18.12 |
0.09228 |
LCAT |
Linoleic acid |
- |
0.65778 |
0.04897 |
0.04897 |
29.1 |
0.05494 |
GAPDH |
Control |
Control |
N/A |
N/A |
N/A |
18.4 |
0.05047 |
LCAT |
Control |
Control |
1 |
0.04774 |
0.04774 |
28.76 |
0.04686 |
Point 6: The resolution of the figure 3 can be improved to be more readable
Response 6: Thank you for reviewer’s comments. We have improved the resolution of Figure 3.
Point 7: Line 316: Fig 8 is mislabelled as Fig 1
Response 7: Thank you for reviewer’s comments. We have updated the corresponding figure in the revised manuscript to reflect the correct description, which is Figure 8A.
Point 8: Figure 2, the volcano plots need the detailed information
Response 8: Thank you for reviewer’s comments. We have added information about the meaning of different colored dots on the volcano plots. Please see Figures 1F-I, 2F-I and 5F-I for details.

Reviewer 2 Report
Dear authors,
Congratulations for your performed work regarding the metabolomics approach to identify the metabolic perturbations in H460 676 cells induced by sodium acetate, sodium butyrate, linoleic acid, and linolenic acid.
Your manuscript is extremely significant, the materials and methods well described, statistics explained, results relevantly exposed and the quality of figures very high.
I do not have any suggestion for improvement.
Kind regards,
Author Response
Response to Reviewer 2 Comments
Dear editor and reviewer,
First of all, we are very grateful for giving us the opportunity to revise our manuscript (molecules-2171777) entitled “Effect of short-chain fatty acids and polyunsaturated fatty acids on metabolites in H460 lung cancer cells”. We appreciate the time and effort that you and the reviewers dedicated to providing insightful and valuable feedback on our manuscript. We have thought carefully about the reviewers’ suggestions and concerns, and the manuscript has been revised accordingly. Our reply is marked in red while the changes made are highlighted in yellow in the manuscript or supporting information and are presented in red in the following response. Please, also note that extra figures and tables containing unpublished data and/or adapted from relevant references were added in this reply only for review to assist in addressing the reviewers’ concerns; these figures and tables are named “figure E” and “table E” followed by a serial number in the order of their appearance and they are not included in the manuscript nor in the supporting information.
Thank you for giving us a positive comments.

Reviewer 3 Report
The authors have done a satisfactory work in presenting role of PUFA and SCFA on lung cancer. I have few suggestions for its plausible publication in the molecules journal.
1. The Abstract is too long, kindly reduce it and write only the few things about background and mainly focus on the results, as same things author already described in the introduction part.
2.Figure 1, image is blurred kindly provide high resolution image.
3.What is the significance value in figure 8.
Author Response
Response to Reviewer 3 Comments
Dear editor and reviewer,
First of all, we are very grateful for giving us the opportunity to revise our manuscript (molecules-2171777) entitled “Effect of short-chain fatty acids and polyunsaturated fatty acids on metabolites in H460 lung cancer cells”. We appreciate the time and effort that you and the reviewers dedicated to providing insightful and valuable feedback on our manuscript. We have thought carefully about the reviewers’ suggestions and concerns, and the manuscript has been revised accordingly. Our reply is marked in red while the changes made are highlighted in yellow in the manuscript or supporting information and are presented in red in the following response. Please, also note that extra figures and tables containing unpublished data and/or adapted from relevant references were added in this reply only for review to assist in addressing the reviewers’ concerns; these figures and tables are named “figure E” and “table E” followed by a serial number in the order of their appearance and they are not included in the manuscript nor in the supporting information.
Point 1: The Abstract is too long, kindly reduce it and write only the few things about background and mainly focus on the results, as same things author already described in the introduction part.
Response 1: Thank you for reviewer’s comments. We have reduced the abstract.
Point 2: Figure 1, image is blurred kindly provide high resolution image.
Response 2: Thank you for reviewer’s comments. We have improved the resolution of Figure 1.
Point 3: What is the significance value in figure 8.
Response 3: Thank you for reviewer’s comments. We have indicated the significance of Figure 8 in both Figure 8A and Figure 8D.

Round 2
Reviewer 1 Report
The authors have provided the sufficient data to support their hypothesis.
Their response to the concerns and queries is satisfactory, thus the revised manuscript can be accepted in the current form.